# Fetal malnutrition and associated factors among term newborn babies in Jimma Zone public hospitals, South West Ethiopia

Aynadis Awoke[1]*, Aynalem Yetwale[2], Tsegaw Biyazin[1], Tegegn Wolde[3], Makeda Sinaga[1]

1 School of Midwifery, Faculty of Health Science, Institute of Health, Jimma University, Jimma, Ethiopia,
2 School of Midwifery, College of Medicine and Health Science, Woldia University, Woldia, Ethiopia,
3 Schools of Midwifery, College of Medicine and Health Science, Wolayta Sodo University, Wolayta Sodo, Ethiopia

* aaynaddis18@gmail.com

## Abstract

### Background

Fetal malnutrition is a major public health burden affecting developing nations, potentially leading to cerebral and neurologic disabilities in later life. Despite its prevalence, little is known about its associated factors in the study area. Thus, this study aimed to assess prevalence and associated factors of fetal malnutrition among term newborn babies in Jimma Zone Public Hospitals.

### Method

A cross-sectional study was carried out among 449-term newborns using systematic sampling techniques in Jimma Zone Public Hospitals from April 1, 2024, to July 30, 2024. Maternal data were collected using an interviewer-administered questionnaire and newborn data were collected using clinical assessment of fetal nutrition (CAN score) scoring system and entered into Epi-data version 4.6 and exported to SPSS version 26 for analysis. Bivariate logistic regression was performed and variables with a p-value ≤0.25 were entered into multivariable logistic regression analysis. P-value less than 0.05 were considered statistically significant and data were presented using text, figures, and tables.

### Result

A total of 449 newborns with their mothers were involved in the study with the response rate of 100%. The prevalence of fetal malnutrition was 91/449 (20.3%) (95% CI, 16.5–24). Among delivered newborns 229/449 (51%) were females and the remaining are males. Maternal age less than nineteen was 60/449 (13.4%) (AOR = 2.930, 95% CI (2.518–13.967)), maternal MUAC ≤23 were 181/449 (40.3%)

**Data availability statement:** All relevant data are within the paper and its Supporting Information files.

**Funding:** The author(s) received no specific funding for this work.

**Competing interests:** No potential conflicts of interest were reported.

**Abbreviations and acronyms:** AGA: Average Gestational Age; ANC: Antenatal Care; BMI: Body Mass Index; CANSCORE: Clinical Assessment of the Nutritional Status; CNS: Central Nervous System; FM: Fetal Malnutrition; IFA: Iron and Folic Acid; IRB: Institutional Review Board; ITN: Insecticide-Treated bed Net; IMC: Integrated Management of Childhood Illness; IPV: Intimate Partner Violence; IUGR: Intrauterine Growth Restriction; JUMC: Jimma University Medical Center; LBW: Low Birth Weight; LDL: Low-Density Lipoprotein; MUAC: Mid-Upper Arm Circumferences; NICU: Neonatal Intensive Care Unit; PI: Ponderal Index; SGA: Small for Gestational Age; SPSS: Statistical Package for Social Science.

(AOR = 4.094, 95% CI (2.155–7.77)), infection during pregnancy 97/449 (21.6%) (AOR = 2.729, 95% CI (1.286–5.792)), malaria 99/449 (22%) (AOR = 2.125, 95% CI (1.002–4.510)), not taking Iron and Folic Acid 300/449 (66.8%) (AOR = 2.897, 95% CI (1.330–6.309)), complication during current pregnancy 116/449 (25.8%) (AOR = 4.629, 95% CI (2.444–8.767)), anemia 301/449 (67%) (AOR = 3.669, 95% CI (1.968–6.840)), low birth weight 77/449 (17.1%) (AOR = 5.363, 95% CI (2.760–10.420)), low placental weight 204/449 (45.4%) (AOR = 4.984, 95% CI (2.530–9.816)), antenatal depression 153/449 (34%) (AOR = 7.184, 95% CI (3.733–13.827)), and intimate partner violence 153/449 (34%) (AOR = 5.613, 95% CI (3.011–10.328)), were significantly associated with fetal malnutrition.

## Conclusion

The prevalence of fetal malnutrition in this study indicates one in five delivered newborn. Newborns with low birth weight, low placental weight, and mothers having anemia, intimate partner violence (IPV), antenatal depression, teenage pregnancy, malaria, infection, and complications during pregnancy were a strong association with fetal malnutrition. Therefore, this study recommends that all concerned bodies, should prioritize efforts to reduce intimate partner violence, prevent infections during pregnancy, enhance maternal nutrition counseling, and address the issue of teenage pregnancy.

## Introduction

Fetal malnutrition is a significant public health problem, affecting one in ten newborns in developed countries and a staggering one in three to one in four in developing countries [1]. It significantly harms newborns, hindering their growth and development, increasing their vulnerability to infections, and raising their risk of chronic diseases later in life. It also contributes to fetal distress in the womb and perinatal complications, including stillbirth, congenital anomalies, neonatal hypoglycemia (low blood sugar), and potentially permanent physical and mental retardation [2].

Fetal malnutrition is a serious concern across India (18.5–21.2%), Nigeria (18.8%), and Ethiopia (21.7%) [3–6]. In Ethiopia, despite efforts like neonatal nurse training and improved healthcare access that have made significant progress towards reducing under-five mortality, newborn deaths remain high (nearly half of under-five deaths) with many preventable causes, and the rate of neonatal mortality still accounts for 41%. This highlights the need for further action to address this critical issue [7].

Studies have shown the devastating impact of malnutrition on various organ systems. It influences endogenous melatonin synthesis, and this effect would be transmitted to the next generation, placing an infant at risk for poor mental performance at a later age. With a concerning 39% of fatally malnourished children exhibiting cerebral and neurological disabilities, infants with fetal malnutrition have significantly higher mortality and morbidity in the first month of life [8,9]. Early identification is significant because intrauterine starvation can lead to later developmental problems [10]. It can cause defects in the formation of the

neural system and also affect brain development. Infants who experience malnutrition in the womb may develop chronic diseases later in life, such as cardiovascular issues, diabetes, and even breast cancer. The fetus's initial adaptations to survive on limited nutrients can have unintended consequences, triggering health problems in adulthood [9,11–13].

In Ethiopia, as well as in sub-Saharan African countries, there is little research conducted about the prevalence of fetal malnutrition and its associated factors, even though there is a high population projection and risk for malnutrition. It is also imperative to achieve the Sustainable Development Goal of newborn health, which aims to reduce neonatal mortality to at least as low as 12 per 1000 live birth [14]. Therefore, this study aimed to address determinants of fetal malnutrition among term newborns delivered in Jimma zone public hospitals.

## Materials and methods

### Study setting and design

The study was conducted in Jimma zone public hospitals, Oromia region, southwest Ethiopia. Jimma zone is one of the twenty administrative zones in the Oromia regional state, which is approximately about 357 kilometers far from Addis Ababa (the capital city of Ethiopia). Jimma zone has three general hospitals, five district hospitals, one referral and teaching Hospital (JUMC), 2 private hospitals, and 120 health centers. It provides service for a total population of 3,486,155 and JUMC serves as a referral hospital for all southwest Ethiopia including south Sudan. A facility-based cross-sectional study was conducted from April 1, 2024, to July 30, 2024.

### Study participants

All newborns with their mothers delivered in Jimma zone public hospitals within 24–48 hours of birth. Selected newborns with their mothers within 24–48 hours of birth delivered in Jimma zone public hospitals were study population.

### Inclusion and exclusion criteria

All newborns with the gestational age of 37–42 weeks and who are singleton live births in Jimma zone public hospitals were included in the study.

This study excluded newborns with obvious congenital abnormalities, incomplete placentas, a requirement for NICU admission; Mothers with known gestational diabetes mellitus, Women who were critically ill and unable to respond to interviews were excluded.

### Sample size determination

The sample size was determined by using a single population proportion formula, considering 21.7% (1) of the prevalence of fetal malnutrition and associated factors among term newborn babies study conducted in Northwestern Ethiopia, with a 4% margin of error at a 95% confidence level.

$$\mathbf{n} = \frac{(Z_{\alpha/2})^2 \, (P)(1-P)}{d^2}$$

$$= \frac{(1.96)^2 (0.217)(1 - 0.217)}{(0.04)^2}$$

no=407.78 =408
nf = 408 + 10%non-respondents.=448.8= 449
Finally, the data was collected from a total sample of 449 newborns with their mothers.

## Sampling technique and procedure

From the nine hospitals found in the Jimma zone, simple random sampling techniques (lottery method) were employed to select 30% from those hospitals, and proportional allocation of the sample was employed based on the monthly delivery report for each selected hospital. Then systematic sampling techniques were employed among mothers who delivered in the selected hospitals based on medical record numbers within 24 to 48 hours. Taking the past year's four-month delivery report of the selected hospital (1650) source population divided by sample size K = 3.67 = 4) the first mother was selected randomly and then every fourth mother was taken (Fig 1.tif).

## Data collection technique and instruments

The data were collected using pre-tested semi-structured interviewer administered questionnaires for maternal data and standardized tools (CAN score) for newborn data [1–7]. The questionnaire consists of seven sections: Socio-demographic characteristics contain nine questions, maternal nutritional and behavioral factors contain seven questions, obstetric factors contain thirteen variables, medical factors contain seven questions, neonatal characteristics have four variables, intimate partner violence (IPV) has thirteen questions, antenatal depression assessment tool contains ten items, and (CAN) score it is a standardized tool, contains nine parameters scored based on loss of subcutaneous fat and muscle mass used to assess the nutritional status of the newborn.

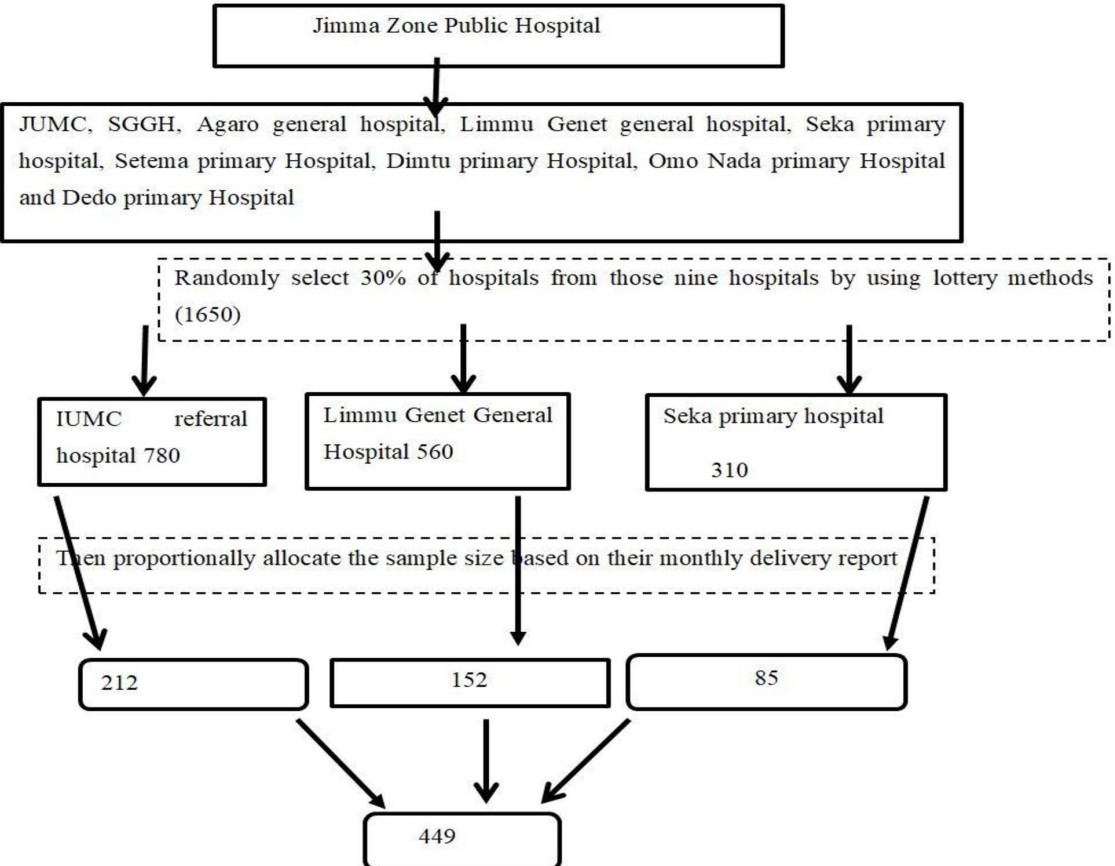

**Fig 1. Schematic presentation of the sampling procedure for fetal malnutrition and associated factors among term newborn babies in Jimma zone public hospital, southwest Ethiopoa.**

## Data collection procedures

The data were collected by six BSc midwives, two for each hospital to do in a turnover period under the supervision of three midwives with MSc degrees. The data was collected by using, face-to-face interview and reviewing medical records, and also physical observation of the newborns nutritional status using CAN score. Following delivery, the placenta was immediately cleared of blood clots. A section approximately 5 cm from the neonatal side was excised and weighed, including membranes and umbilical cord, on an infant weighing scale to the nearest tenth of a gram. The nutritional status of the newborns was assessed using the clinical assessment of nutritional status (CAN) score within 24–48 hours of birth and the rating is based on different characteristics (hair and buccal fat in the cheeks, chin, neck, arms, back inter or subscapular skin, buttocks, legs, chest, and abdominal wall skin). For each point of assessment, the degree of loss of subcutaneous fat was scored by applying a maximum score of four for no evidence of malnutrition and the lowest score of one for the worst evidence of malnutrition. The highest attainable score was thirty-six and the lowest was nine and then the cumulative score is taken to classify whether the fetus is malnourished or well-nourished.

## Dependent variable

**Fetal malnutrition.** Independent Variables

**Socio-demographic characteristics of the mother** (Age, educational status, marital status, Residence, total family size, wealth status, altitude)

**Maternal nutritional and behavioral factors**: -One additional meal during pregnancy, dietary counseling, alcohol consumption, cigarette smoking, chat chawing (sniffing local tobacco), food taboos (restricted food during pregnancy), mid-upper arm circumference of the mother (MUAC)

**Obstetric factors:** -Teenage pregnancy (first conception age), birth interval, parity, wanted and planned pregnancy, previous pregnancy complication, complication in the current pregnancy, antenatal care (ANC) visits, number of ANC contact, Iron and folic acid provision, number of IFA taken, type of complication during pregnancy.

**Medical factors**: -Anemia, infection (UTI or STI), malaria, ITN use, concurrent medical illness, drug use,

**Psychosocial factors**: - Intimate partner violence (IPV) (physical violence, psychological violence, and sexual violence), and antenatal depression.

**Fetal factors**: – Weight of the newborn, sex, and weight of the placenta, gestational age

## Operational definitions

**Clinical Assessment of Nutritional Status (CAN) Score:** -This is a scoring system based on nine superficial readily detectable signs of malnutrition in newborn babies [8].

**Malnourished:** -Indicates undernutrition of the fetus which is the clinical assessment of nutritional status (CAN) cumulative score ≤25(1).

**Well-nourished:-** Indicates which is the clinical assessment nutritional status (CAN) score cumulative score >25.

**Intimate partner violence: –** was assessed as exposure to physical violence (6 items) such as slapping, hitting, kicking, and beating; sexual violence (3 items) including forced sexual intercourse and other forms of sexual coercion; and psychological (emotional) violence (4items) such as insults, belittling and intimidation, threatened to hurt. Women were asked to indicate whether they had experienced any of the violent acts during the current pregnancy and classified as having violent if she had experienced one of these question [9].

**Low placental weight:** – weight of placenta less than 512 gm is considered as low placental weight (1).

**Antenatal Depression**:-Mothers was assessed based on 10 questions or (Edinburgh maternal depression assessment criteria). Those who score greater than 13 are likely to be suffering antenatal depression [10].

## Data quality assurance

Questionnaires were prepared in English first by the principal investigator and translated into the local language (Afan Oromo) and Amharic by another individual who is native to the Afan Oromo language and Amharic respectively. The questionnaire was translated back to English by another individual blinded to the original version of the questionnaire to ensure its consistency. Pre-testing of the data collection tool was done before the actual data collection, in 5% (23) of the total sample size at Agaro General Hospital. Based on the results of pre-testing necessary adjustments to the data collection tools were made, such as language fluency. The training was given to all data collectors and supervisors on the overall procedure of data collection by the principal investigator. During data collection, the supervisors closely follow the day-to-day data collection process and ensure the completeness and consistency of the questionnaire administered each day. After the data was collected, the data was reviewed and checked for completeness before data entry. Besides this, the principal investigator carefully enters and thoroughly cleans the data before the commencement of the analysis.

## Data analysis

The data were cleaned, coded, and entered into Epi data version 4.6. After checking and correcting errors, the data was exported to SPSS version 26 for further analysis. The assumption of binary regression was checked using the chi-square test (cross tabs). The model of fitness was checked by Hosmer and Lemeshow's goodness of fit test and the result (p-value=0.897), indicates the model is a good fit. Bivariate logistic regression analysis was performed for each independent variable with the outcome variable and those variables with a p-value <0.25 were candidates to consider in a multivariable logistic regression. A multivariable logistic regression was performed to control for the effect of confounders of fetal malnutrition. P values less than 0.05 were considered statistically significant and the findings were narrated using text, tables, and figures.

## Ethical approval and informed consent

This study was conducted in accordance with the principles of the Declaration 142 of Helsinki and approved by Jimma University's institutional review board (IRB) with a reference number JUIH/IRB/454/23. Permission letter was provided to, Jimma Zone Health Bureau, and permission from each selected hospital before data collection was taken. Written informed consent and assent for those less than 18 years were obtained from participants after informing, them regarding the purpose, procedures, and benefits of the study and participants were informed that they had the right to refuse or discontinue participating in the research without any compromise in the service they were getting from the respective facilities. They also assured strict confidentiality about any information obtained from them and the information collected was not described concerning individual names.

## Result

The planned sample size was 449, and all of the planned study subjects were involved with a response rate of 100%. In this study, mothers in the age group of ≤19 years or younger were 60/449 (13.4%). Mothers who can read and write were 131/449 (29.2%), regarding occupational status, 188/449 (41.9%) of mothers were employed as housewives, nearly all 397/449 (88.4%) were married and 222/449 (49.4%) followed Muslim religion. More than half 254/449 (56.6%) have families with 4–6 members a little over half 255/449 (56.8%) live in urban areas and also over two-thirds 306/449 (68.2%) live far from a health facility. Regarding their economic status, 87/449 (19.4%) of the respondents were categorized under the poorest household status Table 1.

## Prevalence of fetal malnutrition

In this study, the prevalence of fetal malnutrition was 91/449 (20.3%) which indicates that at least one in five delivered babies identified as malnourished and 358/449 (79.7%) was well-nourished (Fig 2.tif).

**Table 1. Socio-demographic characteristics of the mothers delivered in Jimma zone public hospitals, southwest Ethiopia, 2024 (N = 449).**

| Characteristics | Category | Frequency | Percent (%) |
|---|---|---|---|
| Maternal age | ≤19 | 60 | 13.4 |
| | 20-24 | 129 | 28.7 |
| | 25-29 | 159 | 35.4 |
| | 30-34 | 60 | 13.4 |
| | 35-39 | 33 | 7.3 |
| | ≥40 | 8 | 1.8 |
| Educational status | Unable to read and write | 112 | 24.9 |
| | Able to read and write | 131 | 29.2 |
| | Primary | 35 | 7.8 |
| | Secondary | 60 | 13.4 |
| | College and above | 111 | 24.7 |
| Occupation status | Housewife | 188 | 41.9 |
| | Merchant | 47 | 10.5 |
| | Private worker | 87 | 19.4 |
| | Government worker | 107 | 23.8 |
| | Others[a] | 20 | 4.5 |
| Marital status | Single | 27 | 6 |
| | Married | 397 | 88.4 |
| | Divorced | 25 | 5.6 |
| Religion | Orthodox | 141 | 31.4 |
| | Muslim | 222 | 49.4 |
| | Protestant | 83 | 18.5 |
| | Catholic | 3 | 0.7 |
| Family size | 1-3 | 180 | 40.1 |
| | 4-6 | 254 | 56.6 |
| | ≥7 | 15 | 3.3 |
| Residence | Urban | 255 | 56.8 |
| | Rural | 194 | 43.2 |
| Wealth status | Poorest | 87 | 19.4 |
| | Poor | 97 | 21.6 |
| | Medium | 86 | 19.2 |
| | Rich | 89 | 19.8 |
| | Richest | 90 | 20 |
| Altitude | <2000 | 213 | 47.4 |
| | ≥2000 | 236 | 52.6 |

[a]Others: Include daily labor, maid, and jobless.

## Maternal nutritional and behavioral factors

In this study, 252/449 (56.1%) of mothers reported consuming additional meals during pregnancy, and dietary counseling was received by 282/449 (62.8%) of mothers. During their current pregnancy, mothers 118/449 (26.3%) used alcohol like beer or katikala, while 378/449 (84.2%) did not use cigarettes or local tobacco. Among the mothers, 181/449 (40.3%) were malnourished, defined as having a MUAC of less than 23 cm, and 110/449 (24.5%) reported food taboos within their culture Table 2.

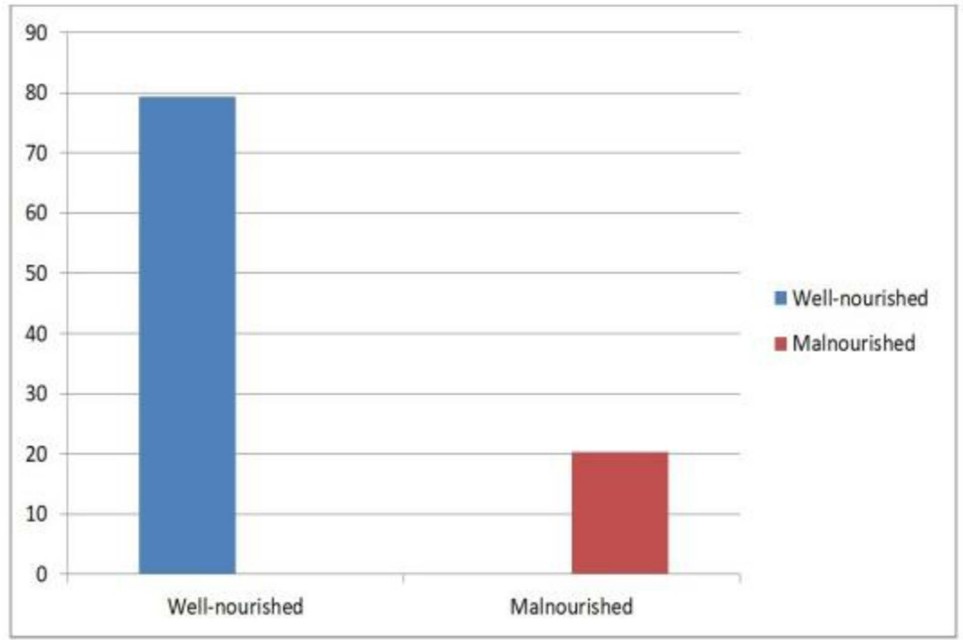

**Fig 2. Prevalence of fetal malnutrition among term newborn babies in Jimma Zone Public Hospital 2024(N=449).**

**Table 2. Maternal nutritional and behavioral factors in Jimma zone public hospitals, southwest Ethiopia, 2024(N=449).**

| Characteristics | Category | Frequency | Percent (%) |
|---|---|---|---|
| Do you consume extra additional meals? | Yes | 252 | 56.1 |
|  | No | 197 | 43.9 |
| Do you get dietary counseling? | Yes | 282 | 62.8 |
|  | No | 167 | 37.2 |
| Have you ever used alcohol, beer, or katikala? | Yes | 118 | 26.3 |
|  | No | 331 | 73.7 |
| Have you ever used cigarettes or local tobacco? | Yes | 71 | 15.8 |
|  | No | 378 | 84.2 |
| Have you ever used Chat chawing? | Yes | 124 | 27.6 |
|  | No | 325 | 72.4 |
| Is there any food item restricted or food taboo in your culture? | Yes | 110 | 24.5 |
|  | No | 339 | 75.5 |
| Maternal MUAC in centimeters | >23 | 268 | 59.7 |
|  | ≤23 | 181 | 40.3 |

## Maternal obstetrics and medical related factors

In this study, mothers with teenage ≤19 years old at their first birth were 185/449 (41.2%) and Primipara comprised 160/449 (35.6%) of the respondents, while grand multiparas accounted for 26/449 (5.8%). Nearly 169/449 (38%) of the mothers had a short birth interval, regarding pregnancy intention, a vast majority 400/449 (89.1%) reported their pregnancies as wanted, and planned pregnancies represented 365/449 (81.7%). Among mothers, 97/449 (21.6%) had an infection during the current pregnancy, and mothers complicated with malaria account for 350/449 (78%). Mothers with comorbid

medical illness were 108/449 (24.1%) of those having a comorbid medical illness, 35/108 (32.4%) have chronic hypertension and 11/108 (10.2%) of mothers were HIV positive (ART users). Mothers who had used drugs such as analgesics during current pregnancy account for 134/449 (29.8%) of the respondents and 301/449 (67%) of the mothers develop anemia during pregnancy Table 3.

### Maternal psychosocial factors

From a history of intimate partner violence, 104/449 (23.2%) experience physical violence, 108/449 (24.1%) experience emotional violence, 131/449 (29.2%) experience sexual violence, and 153/449 (34%) of the mothers experience at least one type of violence. Regarding antenatal depression assessment, 153/449 (34%) of the mothers were categorized as depressed (Fig 3.tif).

### Neonatal characteristics

Among the delivered newborns, 229/449 (51%) were female and 372/449 (82.9%) had normal birth weight. More than half 245/449 (54.6%), of the measured placentas weighed as normal weight, while 204/449 (45.4%) were low placental weight Table 4.

### Associated factors with fetal malnutrition

In this study, a total of thirty five variables were significant factors in bivariate logistic regression at a p-value less than or equal to 0.25 and transfer to multivariable logistic regression.

In multivariable logistic regression, variables such as maternal characteristics like maternal age, place of residence, not gating dietary counseling, adherence to food taboos, MUAC less than 23, presence of infections during pregnancy, malaria, drug use, anemia, newborns with low birth weight, low placental weight, complications during this pregnancy, not taking Iron and Folic Acid (IFA), and psychosocial factors like depression and intimate partner violence, were a significant associations with fetal malnutrition Table 5.

## Discussion

This study aimed to determine the prevalence of fetal malnutrition and its associated factors among term newborns delivered in Jimma Zone public hospitals. The prevalence of fetal malnutrition was 20.3% (95% CI: 16.5–24) which was consistent with other studies finding conducted in Karnataka, India 21.2% [11], Mangalore, India 24% [12] Madhya Pradesh, India 18.5% [13], Uttarakhand, India 17.39% [14], Nepal 18% [5], Portugal 17.5% [15], Ilesha, South-West, Nigeria 18.8% [2] and south Gonder zone, Ethiopia (21.7% [1].

However, the finding of this study is higher than studies conducted in the United States 10.9% [16], Spain 7.6% [17], Port Harcourt, Nigeria 16.7% [4], Lagos, Negeria 14.5% [18], Bhopal, India 8.3% [19], and Debre Markos, Ethiopia 12.32% [7], the discrepancy might be due to multiple factors; America study may be due to difference in food program, socioeconomic between developed country's than in this study under developing countries. The difference between this study and the study conducted in Spanish [17], might be the variation in the study design; the Spanish study employed a follow-up study design, but this study used a cross-sectional study. A discrepancy between our study findings and the findings from the Nigerian study [18] may be the selection criteria. The study conducted in Nigeria focused on term babies delivered at a tertiary hospital and mothers who received antenatal care, however, our study was conducted both on mothers with ANC and without ANC follow-up and also included mothers from primary, general, and referral hospitals.

The prevalence of fetal malnutrition in our study 20.3% was lower than findings reported in other studies: in Turkey 54.8% [20], New Delhi India 27.97% [21], Karnataka, India 52.9% [3], Nigeria 33.9% [22], and Iraq 31% [23]. These discrepancies may be due to several factors such as, The study in Turkey includes preterm newborns, which are known to be

**Table 3. Maternal Obstetrics and medical factors in Jimma Zone public hospital, southwest Ethiopia, 2024(N = 449).**

| Variables | Category | Frequency | Percent (%) |
|---|---|---|---|
| First pregnancy age | ≤19 | 185 | 41.2 |
|  | 20-34 | 257 | 57.2 |
|  | ≥35 | 7 | 1.6 |
| Parity | Primipara | 160 | 35.6 |
|  | Multipara | 263 | 58.6 |
|  | Grand multipara | 26 | 5.8 |
| Birth interval | Optimal birth interval | 280 | 62.4 |
|  | Short birth interval | 169 | 37.6 |
| Is pregnancy wanted | Yes | 400 | 89.1 |
|  | No | 49 | 10.9 |
| Is pregnancy planned | Yes | 367 | 81.7 |
|  | No | 82 | 18.3 |
| Complications in previous pregnancy | Yes | 79 | 17.6 |
|  | No | 370 | 82.4 |
| Type of complication | APH | 20 | 25.3 |
|  | PIH | 11 | 13.9 |
|  | PPH | 11 | 13.9 |
|  | Abortion | 17 | 21.5 |
|  | PROM | 15 | 19.1 |
|  | Others[b] | 5 | 6.3 |
| Complications in this pregnancy | Yes | 116 | 25.8 |
|  | No | 333 | 74.2 |
| Type of complication in this pregnancy | APH | 25 | 21.6 |
|  | PROM | 40 | 34.5 |
|  | Malaria | 20 | 17.2 |
|  | PIH, preeclampsia | 26 | 22.4 |
|  | Others[c] | 5 | 4.3 |
| ANC follow up | Yes | 323 | 71.9 |
|  | No | 126 | 28.1 |
| Number of ANC contact | Regular ANC follow-up | 223 | 67.4 |
|  | Irregular ANC follow-up | 108 | 32.6 |
| IFA supplementation | Yes | 300 | 66.8 |
|  | No | 149 | 33.2 |
| Number of IFA supplementation | A good intake of IFA | 221 | 73.2 |
|  | Poor intake of IFA | 81 | 26.8 |
| Infection during pregnancy | Yes | 97 | 21.6 |
|  | No | 352 | 78.4 |
| Malaria during pregnancy | Yes | 99 | 22.0 |
|  | No | 350 | 78.0 |
| ITN utilization | Yes | 315 | 70.2 |
|  | No | 134 | 29.8 |
| Have you had a comorbid medical illness? | Yes | 108 | 24.1 |
|  | No | 341 | 75.9 |

*(Continued)*

**Table 3.** (Continued)

| Variables | Category | Frequency | Percent (%) |
|---|---|---|---|
| Type of medical illness? | TB | 24 | 22.2 |
| | HIV | 11 | 10.2 |
| | Chronic HTN | 35 | 32.4 |
| | Asthma | 32 | 29.6 |
| | Others[d] | 6 | 5.6 |
| Have you used drugs such as analgesics during pregnancy | Yes | 134 | 29.8 |
| | No | 315 | 70.2 |
| Maternal Hgb | ≤10.9 | 301 | 67.0 |
| | >11 | 148 | 33.0 |

[b]Others include, Infection, Preterm labor, Stillbirth, Anemia [c] Infection (UTI, STI),Hyperemesis gravidraum, Depression/Anxiety, [d] Others: include vascular disease, renal diseases.

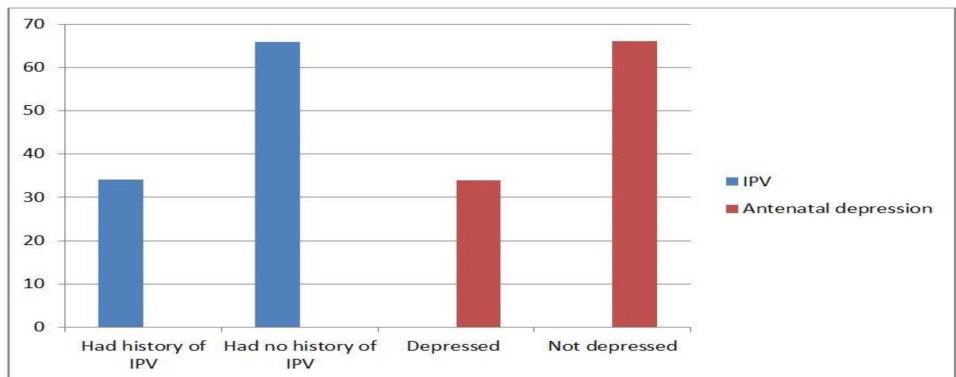

**Fig 3. Intimate partner violence status and antenatal antenatal depression among mothers delivered in Jimma zone public hospital 2024(N=449).**

**Table 4. Neonatal characteristics delivered in Jimma Zone Public Hospital, southwest Ethiopia, 2024 (N = 449).**

| Variables | Category | Frequency | Percent |
|---|---|---|---|
| Sex of newborn | Male | 220 | 49.0 |
| | Female | 229 | 51.0 |
| Weight of newborn | Normal birth weight | 372 | 82.9 |
| | Low birth weight | 77 | 17.1 |
| Placental weight | Normal placental weight | 245 | 54.6 |
| | Low placental weight | 204 | 45.4 |

more susceptible to malnutrition [24]. The difference between our study and the one conducted in New Delhi, India, might be due to the time difference between the studies. Healthcare systems, economic factors, and nutritional landscapes can change over time. The study in Karnataka, India, focused on a rural setting, where limited access to healthcare services and antenatal care during pregnancy can contribute to a higher prevalence of fetal malnutrition [25], however, this study includes both rural and urban areas.

**Table 5. Binary and multivariable logistic regression model, among term newborn babies in Jimma zone public hospital Ethiopia, 2024(N = 449).**

| Characteristics | Category | Malnourished | Well-nourished | COR (95%CI) | AOR (95% CI) | P-value |
|---|---|---|---|---|---|---|
| Age of mother | ≤19 | 31 | 29 | 6.025(3.218-11.28) | 2.930(2.518-13.967) | 0.001 |
| | 20-24 | 8 | 33 | 1.366(0.580-3.218) | 0.924(0.315-2.712) | 0.886 |
| | 25-34 | 33 | 186 | 1 | | |
| | ≥35 | 19 | 110 | 0.974(0.528-1.795) | 0.1859(0.826-4.167) | 0.135 |
| Residence | Urban | 29 | 226 | 1 | | |
| | Rural | 62 | 132 | 3.660(2.241-5.978) | 1.545(1.545-5.511) | 0.018 |
| Dietary counseling | Yes | 23 | 259 | 1 | | |
| | No | 68 | 99 | 7.735(4.569-13.094) | 4.844(2.388-9.828) | 0.001 |
| Food taboo | Yes | 53 | 57 | 7.365(4.45112.188 | 2.095(1.033-4.250) | 0.040 |
| | No | 38 | 301 | 1 | | |
| Maternal MUAC | ≤23 | 64 | 111 | 5.271(3.191-8.718) | 4.094(2.155-7.77) | 0.001 |
| | >23 | 27 | 247 | 1 | | |
| Infection | Yes | 51 | 46 | 8.648(5.157-14.501) | 2.729(1.286-5.792) | 0.009 |
| | No | 40 | 312 | 1 | | |
| Malaria | Yes | 55 | 44 | 10.903(6.447-18.439) | 2.125(1.002-4.510) | 0.049 |
| | No | 36 | 314 | 1 | | |
| Drug use | Yes | 62 | 72 | 8.492(5.094-14.158) | 5.362(2.811-10.228) | 0.001 |
| | No | 29 | 286 | 1 | | |
| Anemia | Yes | 64 | 111 | 5.275(3.191-8.718) | 3.669(1.968-6.8400 | 0.001 |
| | No | 27 | 247 | 1 | | |
| Newborn weight | Low birth weight | 47 | 30 | 11.679(6.700-20.358) | 5.363(2.760-10.420) | 0.001 |
| | Normal birth weight | 44 | 328 | 1 | | |
| Placental weight | Low weight | 75 | 129 | 8.321(4.653-14.881) | 4.984(2.530-9.816) | 0.001 |
| | Normal weight | 16 | 229 | 1 | | |
| Complications in this pregnancy | Yes | 60 | 56 | 10.438(6.212-17.537) | 4.629(2.444-8.767) | 0.001 |
| | No | 31 | 302 | 1 | | |
| Iron Folic Acid taken | Yes | 24 | 276 | 1 | | |
| | No | 67 | 82 | 9.396(5.545-15.922) | 2.897(1.330-6.309) | 0.007 |
| IPV | Yes | 70 | 84 | 10.873(6.302-18.761) | 5.613(3.011-10.328) | 0.001 |
| | No | 21 | 274 | 1 | | |
| Antenatal depression | Depressed | 72 | 81 | 12.959(7.381-22.752) | 5.184(3.733-13.827) | 0.001 |
| | Not depressed | 19 | 277 | 1 | | |

Key: 1 indicates references category, COR: crude odds ratio, AOR: adjusted odds ratio.

In this study, mothers aged less than 19 years were twice more likely to deliver malnourished newborns as compared to mothers who give birth in the age group of 25–29 years old with malnutrition. Supported by studies conducted in India [5], Nepal [14], Nigeria [4] and Debre Markos [7]. A possible justification might be that biologically, younger mothers' bodies may not be fully mature, which can affect the development of the placenta, which plays a critical role in delivering nutrients and oxygen to the fetus [26].They may be more susceptible to nutritional deficiencies due to nutrition and the adolescent transition are closely intertwined, since eating patterns and behaviors are influenced by many factors, including peer influences, parental modeling, food availability, food preferences, cost, convenience, personal and cultural beliefs, mass media, and body image [27].

The current study revealed that, rural residents were more likely to give malnourished newborns as compared to urban residents, this finding aligns with a study's finding in Nigeria [18], and Debre Markos [7]. This variation may be due to several factors. Rural areas often have fewer healthcare providers and limited access to prenatal care, which can hinder the identification and management of nutritional deficiencies during pregnancy [28]. Additionally, there might be lower awareness about the importance of proper nutrition during pregnancy in rural communities, potentially leading to inadequate dietary intake.

In addition, factor significantly associated with a high risk of fetal malnutrition was not receiving dietary counseling. These findings align with previous research conducted in the United States [29], Nigeria [2], and Debre Markos [7]. A possible justification might be that: Dietary counseling during pregnancy can uncover underlying issues contributing to malnutrition risks. This includes eating disorders, and cultural restrictions on certain foods due to pregnancy beliefs. By counseling all pregnant women, these issues can be identified and addressed more effectively promoting better health outcomes for both mother and baby. Dietary counseling goes beyond just addressing malnutrition. It empowers women to make informed choices about their diet, fostering a sense of control and confidence throughout pregnancy and beyond [30].

Furthermore, not taking Iron and Folic Acid (IFA) supplementation during pregnancy is also associated with fetal malnutrition. This finding is supported by studies conducted in the United States, Nigeria, and Debre Markos [2,7,29]. A possible explanation might be that Iron-folic acids (IFA) supplements address deficiencies in specific nutrients and it is crucial for both mother and baby during pregnancy. It prevents anemia that restricts oxygen delivery to the fetus, hindering its growth and development. Furthermore, iron plays a vital role in placental function, promoting proper nutrient and oxygen exchange between mother and fetus. With a low level of iron, the placenta may not function optimally, further contributing to fetal malnutrition [31]. This highlights the importance of taking IFA during pregnancy to prevent fetal malnutrition.

Moreover, other variables statistically associated with fetal malnutrition in this study were, anemia, complications in current pregnancy, and infections during pregnancy. These findings align with studies conducted in Uttarakhand, India, Nepal, and Nigeria [2,3,5,6,13,14]. The possible reasons might be that: Anemia during pregnancy might be a risk for malnourished newborns due to it reduces oxygen and nutrient delivery to the fetus, hinders placental function, and can negatively impact the mother's health [32], all of which can contribute to fetal malnutrition and potentially lead to low birth weight and other health problems for the newborn.

Pregnancy complications were another factor strongly associated with fetal malnutrition. These might be that, first, by increasing nutritional demands: Certain complications can increase the mother's nutrition needs, leaving less available for the fetus. Second, some complications or illnesses can disrupt the body's absorption or utilization of nutrients from food, leading to deficiencies and impairing the placenta's function [33], the critical organ responsible for delivering nutrients and oxygen to the fetus and compromising placental blood flow and nutrient exchange. This limited supply of essential building blocks for growth can lead to fetal malnutrition.

Infection during pregnancy is significantly associated with fetal malnutrition this is supported by studies conducted in India, Nepal, and Nigeria [2,3,13,14]. A possible justification might be that: Infections can cause a decrease in appetite, nausea, and vomiting in the mother. This can lead to inadequate dietary intake and malnutrition in the mother herself [34]. Since the mother is the primary source of nutrition for the fetus, her malnutrition can directly impact fetal growth and development. Also, infections can cause changes in blood flow patterns within the mother's body. This can lead to a decrease in blood flow to the uterus and placenta, limiting the delivery of oxygen and nutrients to the fetus [35]. Early diagnosis, treatment, and close monitoring can lessen infection risks to fetal health.

This study found a significant association between maternal mid-upper arm circumference (MUAC) and fetal malnutrition. Mothers with low MUAC had malnourished newborns than mothers with higher MUAC. This finding aligns with research conducted in India, Nepal, Nigeria, and Debre Markos [2,5,7]. Possible explanations might be that: MUAC is a measure of muscle and fat stores in the mother. A low MUAC often reflects chronic energy deficiency in the mother,

meaning she's not consuming enough calories to meet her needs and those of the developing fetus. This limited energy intake translates to insufficient nutrients reaching the baby, potentially leading to malnutrition [36]. And also low maternal nutritional status may compromise the function of the placenta.

In this study, having a malarial was significantly associated with fetal malnutrition. This finding aligns with previous studies conducted in India, Nigeria, and Debre Markos [2,4,7]. Possible Reasons might be that: Malaria infection can lead to inflammation and nutrient deficiencies, impacting the transfer of essential nutrients from mother to fetus or causing fatigue and loss of appetite, further limiting the mother's nutrient intake and leading to malnutrition in the mother and the developing fetus [37]. Using ITN helps prevent malaria infection during pregnancy, potentially reducing the negative consequences for fetal growth and development. The current study identified drug use during pregnancy, particularly analgesics (pain relievers), as a significant risk factor for fetal malnutrition. This finding aligns with studies conducted in Nigeria and Nepal [2,5]. A possible Reason could be that drug use can disrupt a pregnant woman's appetite and eating habits, leading to her not consuming enough calories and nutrients to support both her and the developing baby. This may lead to placental insufficiency: Drugs can interfere with the placenta's function, which is the organ responsible for transferring nutrients and oxygen from the mother to the fetus [38,39].

This study identified low placental weight is significantly associated with fetal malnutrition. These findings are consistent with previous research conducted in India, Nigeria, Gonder, and Debre Markos [1,2,7,40]. Possible Reasons might be that: Low placental weight can often indicate underlying issues with placental development or function. These issues can involve problems with blood vessel formation, impaired nutrient transport mechanisms, or hormonal imbalances. These can hinder the placenta's ability to nourish the developing fetus, potentially leading to malnutrition. A healthy placenta is a well-developed organ with a large surface area and efficient blood flow. When the placenta is smaller and lighter than expected, its ability to transport these vital elements is compromised. This limited supply of nutrients and oxygen can lead to fetal malnutrition [41].

The finding of this study also shows that fetal malnutrition is five times more common in newborns with low birth weight than newborns with normal birth weight. This finding is consistent with other studies conducted in the United States, Pune, India, Nigeria, and Gonder [1,4,42]. A possible reason could be fetal malnutrition as the cause: in most cases, fetal malnutrition is the underlying cause of low birth weight. When a fetus doesn't receive adequate nutrients and oxygen due to factors like maternal undernutrition, placental problems, or infections, their growth is restricted. This limited growth can lead to low birth weight, indicating the baby may be malnourished. Low birth weight as an Indicator: Low birth weight doesn't necessarily equate to malnutrition. There can be other reasons for a baby to be smaller, such as genetics or premature birth. However, low birth weight, particularly when accompanied by factors like poor muscle tone or thin skin, is a strong indicator that the baby may have experienced some degree of fetal malnutrition [43].

Mothers experiencing intimate partner violence were five times as likely to have malnourished newborns compared to those not experiencing IPV in the current study. This finding is in line with studies conducted in the United States and Gonder [1,44,45]. The possible reasons could be that mothers with experiences of intimate partner violence may be restricted from access to nutritious foods due to financial limitations imposed by the abuser or inability to obtain groceries that limit their dietary choices and also may be limited to access or attend prenatal care appointments [46]. Additionally, some mothers experiencing IPV may turn to drugs or alcohol as a coping mechanism, which has a detrimental impact on the developing fetus, leading to malnutrition and other health problems [47].

In the current study, antenatal depression was a significant factor associated with fetal malnutrition. Mothers with antenatal depression are more likely to deliver malnourished newborns compared to those without depression. This finding is aligning with studies conducted in Australia [48], and Toronto [49]. A possible explanation: Mothers with depression may experience a loss of interest in healthy eating, leading them to skip meals, choose less nutritious options, or neglect their dietary needs. These can limit the essential nutrients reaching the developing fetus [50]. Additionally, depression may make it difficult for some mothers to attend prenatal care appointments, which are crucial for monitoring fetal growth and development. A depressed mother might not have the energy to prepare nutritious meals.

## Strength and limitation of the study

For this study, the CAN score, which relies on physical observation of newborns, may be susceptible to subjectivity. There may be a potential bias in gestational age estimation due to limitations in recalling the last menstrual period, ultrasound accuracy, and rapid assessment techniques. Additionally, there might be social desirability bias in assessing intimate partner violence, where participants may underreport negative experiences.

## Conclusion and recommendation

This study found a high prevalence of malnutrition among newborns delivered in Jimma Zone public hospitals indicating at least one newborn was malnourished among five delivered newborns. Maternal age, maternal residency (rural), not gating dietary counseling, food taboos, low maternal MUAC, not taking Iron and Folic Acid, drug use during pregnancy, anemia, infection, complication during pregnancy, malaria, antenatal depression, intimate partner violence, lower placental weight, were factors associated with fetal malnutrition. Therefore, this study recommended that all concerned bodies should be engaged in mitigate intimate partner violence, prevent infections during pregnancy, enhance maternal nutrition counseling, and address the issue of teenage pregnancy.

## Supporting information

**S1 File. Data set for the study.**
(SAV)

## Acknowledgments

The authors would like to thank all data collectors, study participants, and public health facility administrators/heads.

## Author contributions

**Conceptualization:** Aynadis Awoke, Makeda Sinaga, Aynalem Yetwale, Tsegaw Biyazin, Tegegn Wolde.

**Data curation:** Aynadis Awoke, Makeda Sinaga, Aynalem Yetwale, Tsegaw Biyazin, Tegegn Wolde.

**Formal analysis:** Aynadis Awoke, Makeda Sinaga, Aynalem Yetwale, Tsegaw Biyazin, Tegegn Wolde.

**Funding acquisition:** Aynadis Awoke, Makeda Sinaga, Aynalem Yetwale, Tsegaw Biyazin, Tegegn Wolde.

**Investigation:** Aynadis Awoke, Makeda Sinaga, Aynalem Yetwale, Tsegaw Biyazin, Tegegn Wolde.

**Methodology:** Aynadis Awoke, Makeda Sinaga, Aynalem Yetwale, Tsegaw Biyazin, Tegegn Wolde.

**Project administration:** Aynadis Awoke, Makeda Sinaga, Aynalem Yetwale, Tsegaw Biyazin, Tegegn Wolde.

**Resources:** Aynadis Awoke, Makeda Sinaga, Aynalem Yetwale, Tsegaw Biyazin, Tegegn Wolde.

**Software:** Aynadis Awoke, Makeda Sinaga, Aynalem Yetwale, Tsegaw Biyazin, Tegegn Wolde.

**Supervision:** Aynadis Awoke, Makeda Sinaga, Aynalem Yetwale, Tsegaw Biyazin, Tegegn Wolde.

**Validation:** Aynadis Awoke, Makeda Sinaga, Aynalem Yetwale, Tsegaw Biyazin, Tegegn Wolde.

**Visualization:** Aynadis Awoke, Makeda Sinaga, Aynalem Yetwale, Tsegaw Biyazin, Tegegn Wolde.

**Writing – original draft:** Aynadis Awoke, Makeda Sinaga, Aynalem Yetwale, Tsegaw Biyazin, Tegegn Wolde.

**Writing – review & editing:** Aynadis Awoke, Makeda Sinaga, Aynalem Yetwale, Tsegaw Biyazin, Tegegn Wolde.

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
