## [Decision Letter · Decision Letter 0]

22 Apr 2025

Dear Dr. Awoke,

We look forward to receiving your revised manuscript.

Kind regards,

Tebelay Dilnessa, MSc

Academic Editor

PLOS ONE

Journal Requirements:

Additional Editor Comments:

Use of English language is poor in certain sections and would require a detailed revision.Line 15: Thus, this study aimed to assess predictor……….Lines 17 and 18: A cross-sectional study was carried out among 449-term newborns using systematic sampling techniques in Jimma Zone Public Hospitals from April 1, 2024 to July 30, 2024.In the abstract and result, the absolute number (numerator and denominator) is needed together with the percentage. For example, A/B (C%).Line 72: Materials and methodsLine 80: April 1, 2024 to July 30, 2024.Line 222: (Table 1); remove the word ‘see’; similarly, make a correction for other table and figure citations.Table 6: Follow appropriate scientific notations, for example: this, .924(.315-2.712) should be written as, 0.924(0.315-2.712)Add the following to the declaration section: **Ethical approval and consent to participate, Consent for publication, Data availability statement, Competing interest and Funding statement**
All supplementary files title/description should be written below reference listsThe author should follow uniform font size, font type, paragraphing, etc.Follow the standard binomial nomenclature, italize journal name and the word ‘et al’

Reviewers' comments:

Reviewer's Responses to Questions

**Comments to the Author**

1. Is the manuscript technically sound, and do the data support the conclusions?

Reviewer #1: Yes

2. Has the statistical analysis been performed appropriately and rigorously?

Reviewer #1: Yes

3. Have the authors made all data underlying the findings in their manuscript fully available?

Reviewer #1: Yes

4. Is the manuscript presented in an intelligible fashion and written in standard English?

Reviewer #1: Yes

Reviewer #1: First of all, I would like to appreciate this informative study for the overlooked problem. Fetal malnutrition is currently a public health problem, particularly in developing countries. Such studies are very important for early advocating of maternal well beings. However the authors should address the following issues before publication.

Title: the word “predictors" is not favorable since it was cross-sectional, better to replace with "associated factors".

Abstract: please state separately methods of data collection for maternal data and newborn data (CAN score). Please say p-value less than 0.05. Delete “finally". Please include number of male and female newborns in parentheses. Intimate partner violence and low placental weight needs operational definition in the method section. Please rewrite your conclusion. No need of this study, previous study? Please state your major findings in two or three statements. Even your recommendation is beyond the findings of your study. Please get English language proficiency from native speakers.

Introduction: please revise your introduction particularly the coherence and grammar including punctuations. Dear authors even you missed some published studies, try to include all existing evidences to give best information to the readers.

Methods: dear authors, would you explain why you extend the age of newborns up to 48 hours? Why not within 24 hours?

Please give your suit why 24 - 48 hours post-partum?

Sample size - why you take 0.04?

Who collects the CAN score data? Does the data collectors get CAN score training? Who gave the training?

I am not clear with your independent variables, please clear and simple. Rather it is important to move to the operational definition section.

Result: In your logistic regression table please write only the exact p- values.

How do you classify the placental weight? Have you record the morphology type?

**Do you want your identity to be public for this peer review?** For information about this choice, including consent withdrawal, please see our Privacy Policy

Reviewer #1: **Yes: ** Bickes Wube Sume

---

## [Author Response · Author response to Decision Letter 1]

6 Jul 2025

Response to academic editors and reviewers

Academic Editor1.

Data availability statement

Authors’ response: - all the data are fully available without restriction and it is within the manuscript and its supporting information

Use of English language is poor in certain sections and would require a detailed revision.

• Line 15: Thus, this study aimed to assess predictor……

Authors’ response: Thus, this study aimed to assess prevalence and associated factors of fetal malnutrition among term newborn babies in Jimma Zone Public Hospitals. (Found in line 17&18)

Lines 17 and 18

Authors’ response: A cross-sectional study was carried out among 449-term newborns using systematic sampling techniques in Jimma Zone Public Hospitals from April 1, 2024 to July 30, 2024 (found in line 20&21)

In the abstract and result, the absolute number (numerator and denominator) is needed together with the percentage. For example, A/B (C %)

Authors’ response: Thank you for your feedback we incorporate these comments in the revised manuscript

Line 72: Materials and methods

Authors’ response: Materials and Methods (found in line 82)

Line 80: April 1, 2024 to July 30, 2024

Authors’ response: April 1, 2024 to July 30, 2024 (found in line 90&91.

Line 222: (Table 1); remove the word ‘see’; similarly, make a correction for other table and figure citations.

Table 6: Follow appropriate scientific notations, for example: this, .924(.315 -2.712) should be written as, 0.924(0.315-2.712)

Authors’ response: Thank you for your valuable feedback. We have made the necessary corrections based on your suggestions and have reflected them in the revised manuscript

Add the following to the declaration section: Ethical approval and consent to participate, Consent for publication, Data availability statement, Competing interest and Funding statement

Authors’ response: Thank you for your valuable feedback. We have add those parts in the declaration section in the revised manuscript

Response to reviewer

Reviewer 1

Title: the word “predictors" is not favorable since it was cross-sectional, better to replace with "associated factors

Authors’ response: Dear Reviewer, thank you for your valuable comments and feedback. As per your suggestion, we have replaced the term "predictors" with "associated factors" in the title. Fetal malnutrition and associated factors among term newborn babies in Jimma Zone Public Hospitals, South West Ethiopia

Abstract: please state separately methods of data collection for maternal data and newborn data (CAN score). Please say p-value less than 0.05. Delete “finally". Please include number of male and female newborns in parentheses.

Authors’ response: Under abstract part we add the method of data collection for maternal data is face-to-face interviews and reviewing medical records using semi-structured interviewer administered questionnaire and newborn data is collected by physical observation of fetal nutritional status using standardized tool (CAN score).

We add number of female and male newborn in the parentheses, among delivered newborns 220 (49%) were males and 229 (51%) were females

Please rewrite your conclusion. No need of this study, previous study? Please state your major findings in two or three statements. Even your recommendation is beyond the findings of your study. Please get English language proficiency from native speak

Authors’ response: Conclusion: The prevalence of fetal malnutrition in this study indicates one in five delivered newborn. Newborns with low birth weight, low placental weight, and mothers having anemia, intimate partner violence (IPV), antenatal depression, teenage pregnancy, malaria, infection, and complications during pregnancy were a strong association with fetal malnutrition. Therefore, this study recommended that all concerned bodies should be engaged in mitigate intimate partner violence, prevent infections during pregnancy, enhance maternal nutrition counseling, and address the issue of teenage pregnancy.

Introduction: please revise your introduction particularly the coherence and grammar including punctuations. Dear authors even you missed some published studies, try to include all existing evidences to give best information to the readers.

Authors’ response: We appreciate your comments on the introduction section. We have made efforts to improve the grammar and coherence in the revised manuscript accordingly.

Methods: dear authors, would you explain why you extend the age of newborns up to 48 hours? Why not within 24hours? Please give your suit why 24 - 48 hours post-partum?

Authors’ response: The reason for extending the age of the newborn up to 48 hours was to ensure the inclusion of mothers who delivered via cesarean section, allowing sufficient time for them to stabilize before obtaining their medical history in a comfortable and appropriate condition.

Sample size - why you take 0.04?

Authors’ response: We used a 4% margin of error based on previous research findings confidence interval. In determining the sample size, we considered the p-values and confidence intervals reported in earlier studies. This margin of error was chosen to enhance the precision and reliability of the study results. Selecting an appropriate margin of error is essential for ensuring the accuracy and validity of our statistical estimates.

Who collects the CAN score data? Does the data collectors get CAN score training? Who gave the training?

Authors’ response: The newborn data were collected by BSc midwives under the supervision of MSc midwives who has work experiences in neonatal side. Prior to data collection, the principal investigators provided training on the use of the CAN score, including guidance on how to assess and assign scores based on the loss of fat and muscle mass across various physical characteristics of the newborn

Intimate partner violence and low placental weight needs operational definition in the method section

Authors’ response: Dear academic editors, Thank you for your valuable feedback in operational definition of these variables, in the revised manuscript we put the operational definitions accordingly

How do you classify the placental weight? Have you record the morphology type?

Authors’ response: We classified the placenta after measuring its weight using newborn weighting scale measurement, thickness, and length by using tape meter. The classification was based on placental weight, distinguishing between low placental weight and normal placental weight, following the criteria established by previous research studies, as referenced in the manuscript

I am not clear with your independent variables, please clear and simple. Rather it is important to move to the operational definition section.

Authors’ response: Our independent variables include various maternal characteristics, categorized into the following factors: socio-demographic factors, maternal nutritional and behavioral factors, obstetric factors, medical factors, psychosocial factors, and fetal factors. The fetal factors include the weight of the newborn, sex, weight of the placenta, and gestational age. We have listed the specific variables under each category.

Result: In your logistic regression table please write only the exact p- values.

Authors’ response: We appreciate your comment regarding the logistic regression table. Accordingly, we have made the necessary revisions as per your suggestion

---

## [Decision Letter · Decision Letter 1]

18 Jul 2025

Dear Dr. Awoke,

Thank you for submitting your manuscript to PLOS ONE. After careful consideration, we feel that it has merit but does not fully meet PLOS ONE’s publication criteria as it currently stands. Therefore, we invite you to submit a revised version of the manuscript that addresses the points raised during the review process.

We look forward to receiving your revised manuscript.

Kind regards,

Tebelay Dilnessa, MSc

Academic Editor

PLOS ONE

Journal Requirements:

Additional Editor Comments:

Table 6: P=0.000, What does it mean.The author should make it meaningful.Table 6: Why you consider as a reference age group from 25-34? Do you have reason?The denominator still should be included in the percentage of your both in the abstract and main result.Some of the tables can be integrated and prepared. Think of it.

Reviewers' comments:

Reviewer's Responses to Questions

**Comments to the Author**

Reviewer #1: All comments have been addressed

2. Is the manuscript technically sound, and do the data support the conclusions?

Reviewer #1: Yes

3. Has the statistical analysis been performed appropriately and rigorously?

Reviewer #1: Yes

4. Have the authors made all data underlying the findings in their manuscript fully available?

Reviewer #1: Yes

5. Is the manuscript presented in an intelligible fashion and written in standard English?

Reviewer #1: Yes

Reviewer #1: the findings of this study are very important to prevent neonatal life sequelae due to fetal malnutrition, and informative for the ANC services . All my comments were fully addressed, I think it is feasible for publication.

**Do you want your identity to be public for this peer review?** For information about this choice, including consent withdrawal, please see our Privacy Policy

Reviewer #1: **Yes: ** Bickes Wube Sume

---

## [Author Response · Author response to Decision Letter 2]

28 Aug 2025

Response to academic editors and reviewers

First of all I would like to thank you for giving us the opportunity to submit a revised draft of the manuscript “Fetal malnutrition and associated factors among term newborn babies in Jimma Zone Public Hospitals, South West Ethiopia” we appreciate the time and effort that you and the reviewers dedicated to providing feedback on our manuscript and are grateful for the insightful comments on and valuable improvement to our paper. We have incorporated most of the suggestions made by the academic editors and reviewers. Those changes are highlighted in the revised manuscript with track changes. Please see below, in blue, for a point- by-point response to the academic editors and reviewers comments and concerns.

Academic Editor1.

Table 6: P=0.000, what does it mean. The author should make it meaningful.

Authors’ response: - Thank you for your valuable feedback. We agree with your point, and in the revised manuscript we have approximated this number to 0.001 for clarity and consistency

Table 6: Why you consider as a reference age group from 25-34? Do you have reason?

Authors’ response: - Thank you for raising this important point. The comment is correct, and we agree with the observation. The reason for considering the age group 25–34 years as the reference category is that it represents the biological optimum and lowest-risk reproductive age group. Women in this age bracket are often at their peak reproductive potential, with high fertility and relatively lower risks of pregnancy complications compared to younger mothers (<20 years) and older mothers (≥35 years).

For studying risk factors such as fetal malnutrition, it is necessary to identify a ‘baseline’ group for comparison. The 25–34 age range provides this baseline because it reflects the most physiologically stable and nutritionally favorable stage, thereby minimizing the influence of age-related risks. Furthermore, evidence from epidemiological studies consistently demonstrates a U-shaped relationship between maternal age and adverse outcomes: increased risks at the extremes (<20 and ≥35 years), with the lowest risk observed in the middle group (25–34 years). For these reasons, this age group is commonly used as the reference category in similar studies.

The denominator still should be included in the percentage of your both in the abstract and main result

Authors’ response: - Thank you for this valuable comment. In the revised manuscript, we have included both the denominators and the corresponding percentages in the relevant sections.

Some of the tables can be integrated and prepared. Think of it.

Authors’ response: - Thank you for your observation. Yes, the tables have been integrated. Specifically, we have combined Table 3 and Table 4, which presented maternal obstetric and medical factors, into a single table in the revised manuscript

If the reviewer comments include a recommendation to cite specific previously published works, please review and evaluate these publications to determine whether they are relevant and should be cited

Authors’ response: -Thank you for this constructive comment. Although the reviewer did not recommend specific previously published works, the suggestion to check for uncited relevant literature was very helpful. We carefully reviewed the existing body of evidence, and most of the relevant studies have now been incorporated

Authors’ response: - Thank you for your observation. We have carefully reviewed the entire reference list to ensure its completeness and accuracy, and the necessary corrections have been made. These updates have been highlighted in the revised manuscript.

---

## [Editor Report · Decision Letter 2]

2 Sep 2025

Fetal Malnutrition and Associated Factors Among Term Newborn Babies in Jimma Zone Public Hospitals, Southwest Ethiopia

PONE-D-24-57243R2

Dear Dr. Awoke,

We’re pleased to inform you that your manuscript has been judged scientifically suitable for publication and will be formally accepted for publication once it meets all outstanding technical requirements.

Kind regards,

Tebelay Dilnessa, MSc

Academic Editor

PLOS ONE
---

## [Editor Report · Acceptance letter]

PONE-D-24-57243R2

PLOS ONE

Dear Dr. Awoke,

I'm pleased to inform you that your manuscript has been deemed suitable for publication in PLOS ONE. Congratulations! Your manuscript is now being handed over to our production team.

Kind regards,

on behalf of

Dr. Tebelay Dilnessa

Academic Editor

PLOS ONE